# Association between Energy Balance-Related Factors and Clinical Outcomes in Patients with Ovarian Cancer: A Systematic Review and Meta-Analysis

**DOI:** 10.3390/cancers14194567

**Published:** 2022-09-20

**Authors:** Stephanie Stelten, Christelle Schofield, Yvonne A. W. Hartman, Pedro Lopez, Gemma G. Kenter, Robert U. Newton, Daniel A. Galvão, Meeke Hoedjes, Dennis R. Taaffe, Luc R. C. W. van Lonkhuijzen, Carolyn McIntyre, Laurien M. Buffart

**Affiliations:** 1Department of Physiology, Radboud Institute of Health Sciences, Radboud University Medical Center, 6525 GA Nijmegen, The Netherlands; 2Exercise Medicine Research Institute, Edith Cowan University, Perth 6027, Australia; 3Department of Obstetrics and Gyneacology, Center for Gynaecologic Oncology Amsterdam (CGOA), Amsterdam UMC, University of Amsterdam, 1105 AZ Amsterdam, The Netherlands; 4Department of Gynecology, Center for Gynecologic Oncology Amsterdam (CGOA), The Netherlands Cancer Institute–Antoni van Leeuwenhoek Hospital, 1066 CX Amsterdam, The Netherlands; 5Department of Obstetrics and Gynecology, Center for Gynecologic Oncology Amsterdam (CGOA), Cancer Center Amsterdam, Amsterdam UMC, Vrije Universiteit Amsterdam, 1081 HV Amsterdam, The Netherlands; 6Department of Medical and Clinical Psychology, CoRPS-Center of Research on Psychological and Somatic Disorders, Tilburg University, 5000 LE Tilburg, The Netherlands

**Keywords:** body composition, diet, exercise, ovarian cancer, meta-analysis

## Abstract

**Simple Summary:**

Ovarian cancer and its treatment are associated with energy balance-related problems, such as overweight, malnourishment, compromised muscle mass and quality, and physical inactivity. This may impact the quality of life and treatment outcome. These factors may be modifiable, and women with ovarian cancer have indicated that they want to do something themselves to help improve their treatment outcome. In order to better understand the role of energy-balance-related problems in patients treated for ovarian cancer, this study synthesized the available research on (i) the association of body weight, body composition, diet, and physical activity or exercise with survival or treatment-related complications and (ii) the evidence from exercise- and/or dietary interventions. The results indicate that body mass index has a limited prognostic value, while other measures of body composition may have more prognostic potential. Additionally, the findings provide important leads for future research directions.

**Abstract:**

Background: This systematic review and meta-analysis synthesized evidence in patients with ovarian cancer at diagnosis and/or during first-line treatment on; (i) the association of body weight, body composition, diet, exercise, sedentary behavior, or physical fitness with clinical outcomes; and (ii) the effect of exercise and/or dietary interventions. Methods: Risk of bias assessments and best-evidence syntheses were completed. Meta-analyses were performed when ≥3 papers presented point estimates and variability measures of associations or effects. Results: Body mass index (BMI) at diagnosis was not significantly associated with survival. Although the following trends were not supported by the best-evidence syntheses, the meta-analyses revealed that a higher BMI was associated with a higher risk of post-surgical complications (*n* = 5, HR: 1.63, 95% CI: 1.06–2.51, *p* = 0.030), a higher muscle mass was associated with a better progression-free survival (*n* = 3, HR: 1.41, 95% CI: 1.04–1.91, *p* = 0.030) and a higher muscle density was associated with a better overall survival (*n* = 3, HR: 2.12, 95% CI: 1.62–2.79, *p* < 0.001). Muscle measures were not significantly associated with surgical or chemotherapy-related outcomes. Conclusions: The prognostic value of baseline BMI for clinical outcomes is limited, but muscle mass and density may have more prognostic potential. High-quality studies with comprehensive reporting of results are required to improve our understanding of the prognostic value of body composition measures for clinical outcomes. Systematic review registration number: PROSPERO identifier CRD42020163058.

## 1. Introduction

Ovarian cancer is mostly diagnosed at an older age [1] and at an advanced stage according to the International Federation of Gynecology and Obstetrics (FIGO) [2]. Patients with ovarian cancer often face energy balance-related problems such as overweight and obesity [3,4,5], malnourishment, and compromised skeletal muscle mass and density [6]. This may increase their risk of poorer treatment outcomes including post-surgical complications [7,8,9], shorter time to disease progression [10,11,12], and all-cause mortality [9,12,13]. Additionally, most patients with ovarian cancer have reduced physical activity levels after diagnosis and remain insufficiently active during and after treatment [14]. Higher physical activity and a healthier body weight have been demonstrated to be related to a higher quality of life [14,15] and physical function [16] in patients with ovarian cancer. However, the effects of malnourishment and an unhealthier body composition on patient-reported outcomes is not well understood in this cancer population. These energy balance-related concerns are modifiable, and women with ovarian cancer have indicated that they want to do something themselves to help improve their treatment outcome [17].

The role of age, comorbidities, and cancer-related characteristics such as tumor stage, histology, and extent of surgery on clinical outcomes is well documented [18,19,20,21,22,23]. However, the association of modifiable factors such as body weight, body composition, diet, exercise, and sedentary behavior with survival and treatment-related outcomes in patients with ovarian cancer has not yet been fully elucidated. Research findings on the association of body composition with clinical outcomes in patients with ovarian cancer are often ambiguous or contradictory [8,12,24,25,26,27,28,29], while little is known about the association of post-diagnosis exercise and dietary behavior with clinical outcomes [30]. Additionally, while there is substantial evidence that exercise and/or dietary interventions are effective to maintain or improve physical activity and fitness, body composition, and quality of life in patients with other types of cancer, such as breast and prostate cancer [31,32], there is limited information available in patients with ovarian cancer during treatment [14,33,34]. Moreover, the effects of such interventions on clinical outcomes are unknown.

A better understanding of the association between modifiable energy balance-related factors and clinical outcomes in ovarian cancer patients will inform appropriate and timely assessment and the design and implementation of ovarian cancer-specific exercise and/or dietary interventions in research and clinical settings. Therefore, the purpose of this review and meta-analysis was to synthesize current evidence on the association of body weight, body composition, diet, exercise, sedentary behavior, and physical fitness at diagnosis and during treatment with clinical outcomes in patients with ovarian cancer. Furthermore, we aimed to summarize evidence on the effect of exercise and/or dietary interventions during treatment in patients with ovarian cancer.

## 2. Materials and Methods

### 2.1. Search Strategy and Study Selection

For this study, we performed two systematic searches. First, we searched for observational studies examining the association of body weight, body composition (i.e., body mass index (BMI), fat mass, muscle mass and/or muscle density), diet, exercise, sedentary behavior, or physical fitness at diagnosis and/or during first-line cancer treatment with survival and treatment-related outcomes in patients with ovarian cancer. Second, we searched for experimental studies examining the effect of an exercise and/or dietary intervention delivered during first-line treatment on body weight, body composition, dietary intake, physical activity, biomarkers, and patient-reported outcomes or survival and treatment-related outcomes in patients with ovarian cancer. An overview of the inclusion and exclusion criteria per systematic search is presented in Table 1. From studies with nearly identical datasets, the most relevant study was selected for inclusion.

The searches were conducted in the PubMed, EMBASE, PsycINFO, Cochrane Library, SPORTDiscus, and CINAHL databases for peer-reviewed published studies up to November 2021. Keywords related to ovarian cancer, body weight, body composition, diet, physical activity, exercise, sedentary behavior, physical fitness, and lifestyle were used. An example of the search conducted in PubMed can be found in Table 2. Additionally, a manual search was undertaken in the reference lists of relevant review papers. After removing duplicates, the titles and abstracts were independently screened by two reviewers (S.S., C.S.) using the Rayyan platform [35]. Subsequently, full text articles were assessed for eligibility by the same two reviewers. Reviewers were blinded to each other’s decisions. Disagreements and uncertainties were resolved by discussion with a third and fourth reviewer (L.B., C.M.). All procedures undertaken in this systematic review and meta-analysis were reported in accordance with the Cochrane Back Review Group [36] and the Preferred Reporting Items for Systematic Reviews and Meta-Analysis statement [37]. The protocol has been registered in the International Prospective Register of Systematic Reviews (PROSPERO identifier: CRD42020163058).

### 2.2. Data Extraction 

Data extraction was performed independently by two reviewers (S.S. and C.S. for observational studies, and S.S. and Y.H. for experimental studies) using standardized forms. For all studies, details including the country of origin, sample size, age, cancer stage, cancer treatment, timing, location, and methods of assessments, and follow-up period were extracted, as well as hazard ratios (HR) from studies investigating the association of body composition or body weight measures with overall or progression-free survival, and odds ratios (OR) from studies investigating the association between body weight measures and post-surgical complications with their associated measures of variability such as 95% confidence intervals (CI) or standard errors when available. Furthermore, for experimental studies, information about the intervention and control arms was extracted.

### 2.3. Risk of Bias 

The risk of bias was assessed independently by two reviewers using the Joanna Briggs Institute Critical Appraisal tool [38] for observational studies (S.S. and C.S.) and the Cochrane risk-of-bias tool for experimental studies (S.S. and Y.H.). The Joanna Briggs Institute Critical Appraisal tool consists of eleven items related to study design, conduct, and analysis. Studies were rated as having low, high, unclear, or not applicable risk of bias in the following items: (1) clear inclusion and exclusion criteria; (2) measurement of exposure; (3) method of measurement of exposure; (4) confounding factors; (5) strategies to deal with confounding factors; (6) free of outcome at start of the study; (7) measurement of outcome; (8) follow-up time; (9) completeness of follow-up; (10) strategies for managing incomplete follow-up; and (11) statistical analysis. Low risk-of-bias papers were defined by ≥7 positive answers, moderate risk-of-bias by 4–6 positive answers, and high risk-of-bias by 1–3 positive answers [39]. The Cochrane risk-of-bias tool 2.0 includes judgments of low or high risk of bias, or some concerns of bias for the following items: (1) randomization process; (2) deviations from the intended intervention (i.e., effect of assignment to intervention or effect of adhering to intervention); (3) missing outcome data; (4) measurement of outcome; and (5) selective reporting [40]. Disagreements were resolved by consensus in discussion with two other reviewers (L.B., C.M.).

### 2.4. Best-Evidence Synthesis and Meta-Analysis 

A best-evidence synthesis was applied in which the number of studies, risk of bias, and consistency of study results were considered. The evidence level was rated as follows: (A) strong evidence when there were consistent findings in ≥2 studies with a low risk of bias; (B) moderate evidence when there were consistent findings in one study with a low risk of bias and ≥1 study with a high risk of bias, or in ≥2 studies with a high risk of bias; or (C) insufficient evidence when there were inconsistent findings in ≥2 studies (C1) or when only one study was available (C2) [41]. Results were considered consistent when ≥75% of the studies showed results in the same direction. Different results for ovarian cancer subgroups in the same study were not considered as inconsistent. 

Meta-analyses were performed if estimates and measures of variability of associations or effects were reported in at least three papers. HRs and ORs were extracted from multivariable models and log-transformed to be included in separate meta-analysis models. Data were pooled using inverse variance random-effects models. A *p*-value of ≤0.05 was considered statistically significant. Forest plots were generated to illustrate the main results. Heterogeneity between studies was tested using the I^2^ statistic and the *p*-value from the χ2-based Cochran’s Q test with a high heterogeneity defined by a threshold *p*-value of 0.1 or I^2^ value greater than 50% [42]. Outliers were examined using sensitivity analysis by omitting one study at a time. To check for publication bias, contour-enhanced funnel plots of log HR or OR against their standard error were generated and explored using Egger’s regression asymmetry test when more than ten studies were available [43]. Analyses were conducted using the Review Manager (RevMan) software version 5.4, from the Cochrane Collaboration 2020 (Copenhagen: The Nordic Cochrane Centre) and the package ‘meta’ from R (R Core Team, 2020).

## 3. Results

### 3.1. Study Selection

In total, 5423 observational studies and 3736 experimental studies were identified. After removing duplicates and screening titles and abstracts, 186 observational and 83 experimental studies were eligible for full-text screening. In total, 73 observational and 4 experimental studies were eligible for inclusion in this systematic review. A total of 25 observational studies were eligible and included in the meta-analyses (Figure 1).

### 3.2. Observational Studies

The included observational studies examined the association of body weight, body composition, diet, or physical fitness with clinical outcomes (Table 3). No observational studies on exercise or sedentary behavior were found. A retrospective study design was used for all but three included studies [44,45,46]. Patients with FIGO stage III-IV were included in 39 studies, 30 studies included patients with all stages, 2 studies included FIGO stage I-II, and stage was not specified in 2 other studies. In total, 34 studies included only patients who had received primary cytoreductive surgery and adjuvant chemotherapy, 8 studies included only patients who had received neoadjuvant chemotherapy and interval cytoreductive surgery, 21 studies included patients on both treatment regimens, and the order of surgery and chemotherapy was unclear for 10 studies. 

Most studies (82.5%) reported body mass index (BMI) using categories recommended by the World Health Organization [47], with a BMI < 18.5 kg/m^2^ classified as underweight; 18.5–24.9 kg/m^2^ as normal weight; 25.0–29.9 kg/m^2^ as overweight; and ≥30.0 kg/m^2^ as obese. The remaining studies [10,24,44,48,49,50,51,52,53,54] used various BMI categories recommended for Asian or Western Pacific populations. A total of 25 studies investigated measures of muscle mass, muscle density, and/or fat mass using computed tomography (CT) scans routinely conducted for diagnostic or surveillance purposes. Most studies measured muscle mass as the total abdominal muscle cross-sectional area at the third lumbar vertebral level normalized for height to determine skeletal muscle index (SMI, cm^2^/m^2^), muscle density as the average Hounsfield Units (HU) of the total abdominal muscle area on the selected image(s), and fat mass in cm^2^ as the total fat area, subcutaneous fat area, and/or visceral fat area. Two separate studies reported on the association of diet [55] and physical fitness [56] with clinical outcomes. Most observational studies (84%) had a low risk of bias (Table 4; complete risk-of-bias assessment). 

**Table 3 cancers-14-04567-t003:** Descriptive characteristics of 73 observational and 4 experimental studies.

Observational Studies								
Author Year	Country	Sample Size	Age (Years) (±SD or Range)	FIGO Stage (% of Patients)	Treatment (% of Patients)	Risk of Bias Assessment	Determinant	Outcome
Ansell 1993 [57]	South Africa	127	Median: 58	IIIB-IV EOC	PDS followed by chemotherapy	Low	Weight change	− Overall survival
Ataseven 2018 [58]	Germany	323	Median: 60 (21–89)	IIIB-IV EOC	PDS	Low	Muscle density Muscle mass	− Overall survival
Aust 2015 [59]	Austria	140	Mean: 60 ± 13	I-IV EOC	PDS followed by chemotherapy	Low	BMI Muscle density Muscle mass	− Overall survival − Progression-free survival
Bacalbasa 2020 [60]	Romania	80	Median: 52.6 (24–83)	IIIC-IV EOC	PDS followed by chemotherapy (91.3%), NACT-IDS (8.7%)	Moderate	BMI	− Post-surgical complications
Backes 2011 [61]	USA	187	Mean: BMI < 25 = 57.2 ± 12.5 BMI 25–30 = 59.3 ± 9.7 BMI > 30 = 58.6 ± 8.8	III-IV EOC, primary peritoneal or fallopian tube cancer	PDS followed by chemotherapy	Low	BMI	− Overall survival − Progression-free survival
Bae 2014 [24]	Korea	236	Mean: BMI < 18.5 = 49 (29–76) BMI 18.5–22.9 = 51 (13–79) BMI 23–24.9 = 65 (24–76) BMI 25–29.9 = 69 (38–78) BMI ≥ 30 = 54 (35–76)	III-IV EOC	PDS followed by chemotherapy (98.3%), NACT-IDS (1.7%)	Low	BMI	− Overall survival
Barrett 2008 [62]	Scotland	1077 (survival analysis for 1067)	Median: 59 (19–85)	IC-IV OC or primary peritoneal cancer	PDS followed by chemotherapy (docetaxel-carboplatin, N = 537, or paclitaxel-carboplatin, N = 538)	Moderate	BMI	− Extent of debulking surgery − Overall survival − Progression-free survival − Toxicity-induced modification of treatment
Bronger 2017 [63]	Germany	128	Median: 65 (33–85)	III-IV EOC	PDS followed by chemotherapy	Low	BMI Muscle mass and change	− Overall survival
Bruno 2021 [64]	Brazil	239	Mean: 56.3 ± 11.4	I-IV EOC	Chemotherapy	Low	Fat mass Muscle density Muscle mass	− Chemotherapy toxicity − Overall survival
Califano 2013 [65]	Italy	117 (BMI unknown for 10.3%)	Median: 56 (59–84)	I-II (9.4%), III-IV (90.6%) OC	PDS followed by chemotherapy	Low	BMI	− Chemotherapy response − Overall survival − Progression-free survival
Castro 2018 [20]	Brazil	83 (BMI unknown for 1.2%)	69.9% = ≤60 30.1% = >60	III-IV OC	PDS followed by chemotherapy (51.8%), NACT-IDS (48.2%)	Low	BMI	− Post-surgical complications − Toxicity-induced modification of treatment
Chae 2021 [66]	Korea	82	Median: 52 (18–83)	I-II OC	PDS followed by chemotherapy (91.5%), NACT-IDS (8.5%)	Low	Muscle mass	− Disease-free survival − Overall survival
Chokshi 2022 [67]	USA	90	Mean: 63.13 ± 12.33	III-IV OC, primary peritoneal or fallopian tube cancer	NACT	Moderate	BMI	− Chemotherapy complications
Conrad 2018 [68]	USA	102	Mean: 55 ± 11	III-IV EOC, primary peritoneal or fallopian tube cancer	PDS followed by chemotherapy	Low	Fat mass Muscle mass	− Chemotherapy toxicity − ICU admission − Length of hospital stay − Overall survival − Post-surgical complications − Progression-free survival − Toxicity-induced modification of treatment
Davis 2016 [69]	USA	92	Mean: BMI 18.5–24.9 = 58.7 BMI 25–29.9 = 55.8 BMI ≥ 30 = 59.4	IIIC EOC, primary peritoneal or fallopian tube cancer	PDS followed by (intraperitoneal) chemotherapy	Low	BMI	− Chemotherapy complications − Chemotherapy response − Overall survival − Platinum disease-free survival − Platinum sensitivity − Progression-free survival − Toxicity-induced modification of treatment
Di Donato 2021 [70]	Italy	263	Mean: 55.2 ± 12.5	III-IV OC	PDS followed by chemotherapy (61.2%), NACT-IDS (38.8%)	Low	BMI	− Post-surgical complications
Duska 2015 [18]	USA	1873	Patient not re-hospitalized = 59.8 Patients re-hospitalized = 62	III-IV EOC, primary peritoneal or fallopian tube cancer	PDS followed by chemotherapy with or without BEV (NR)	Low	BMI	− Re-hospitalization
Element 2022 [56]	UK	43	Mean: Low VO_2_ max 68.34 ± 4.36 Normal VO_2_ max 61.76 ± 5.41	III-IV OC	PDS followed by chemotherapy (N = 17), NACT-IDS (N = 26)	Low	VO_2_ max Anaerobic threshold	− Extent of debulking surgery − Overall survival − Post-surgical complications
Fotopoulou 2011 [71]	Germany	306	Median: 58 (18–92)	I-IV EOC	PDS	Low	BMI	− Extent of debulking surgery − Overall survival − Post-surgical complications − Progression-free survival
Hanna 2013 [72]	USA	325 (BMI unknown for 9.8%)	Median: 60 (24–84)	III-IV EOC	PDS followed by chemotherapy	Low	BMI	− Overall survival − Progression-free survival − Toxicity-induced modification of treatment
Hawarden 2021 [73]	UK	208	Median: Survival < 100 days = 73 (37–84), Survival > 100 days = 67 (37–90)	I-IV OC	PDS followed by chemotherapy, NACT-IDS, best supportive care	Low	BMI	− Overall survival
Hess 2007 [74]	USA	645	44.3% = <55 28.5% = 55–64 27.2% = ≥65	III EOC	PDS followed by chemotherapy	Low	Weight change	− Overall survival − Progression-free survival
Heus 2021 [75]	Netherlands	298	Mean: 62 (21–91)	III-IV OC	PDS followed by chemotherapy, NACT-IDS (75.8%)	Low	Fat mass Muscle mass	− Post-surgical complications
Hew 2014 [76]	USA	370	Mean: BMI < 30 = 58.2 ± 12.2 BMI ≥ 30 = 57.3 ± 10.5	I-II (39.2%), III-IV (59.2%), unstaged (1.6%) EOC	PDS followed by chemotherapy	Low	BMI	− Progression-free survival − Recurrence-free survival
Huang 2020 [11]	Taiwan	139	Mean: 54.4 ± 10.3	III EOC	PDS followed by chemotherapy	Low	Fat mass and change Muscle density and change Muscle mass and change	− Overall survival − Progression-free survival
Inci 2021 [77]	Germany	106	Median: 57 (18–87)	I-IV OC	PDS followed by chemotherapy, NACT-IDS (N = 11)	Low	BMI	− Post-surgical complications
Jiang 2019 [48]	China	160	Median: 54 (28–73)	III-IV EOC, primary peritoneal or fallopian tube cancer	NACT-IDS	Low	BMI	− Extent of debulking surgery
Kanbergs 2020 [78]	USA	507	Mean: BMI ≥ 30 + NACT = 63.8 ± 9.5, BMI ≥ 30 + PDS = 61.8 ± 9.4 BMI < 30 + NACT 63.7 ± 10.6 BMI < 30 + PDS = 61.7 ± 10.8	IIIC-IV EOV, primary peritoneal or fallopian tube cancer	NACT-IDS	Low	BMI	− Post-surgical complications − Re-hospitalization − Toxicity-induced modification of treatment
Kim 2014 [49]	Korea	360	Mean: 53.9 (18–80)	III-IV EOC, primary peritoneal or fallopian tube cancer	PDS followed by chemotherapy (84.2%), NACT-IDS 15.8%	Low	BMI and change	− Overall survival − Progression-free survival
Kim 2020 [50]	Korea	179	Mean: 57.5 ± 11.3	III-IV OC	PDS followed by chemotherapy (75.4%), NACT-IDS (24.6%)	Low	BMI Fat mass Muscle mass	− Overall survival − Progression-free survival
Kim 2021 [51]	Korea	208	Mean: 54.4 ± 10.7	I-IV OC, primary peritoneal or fallopian tube cancer	PDS followed by chemotherapy (82.2%), NACT-IDS (17.8%)	Low	BMI and change Fat mass and change Muscle mass and change	− Overall survival − Progression-free survival
Kumar 2014 [4]	USA	620	Mean: 64.6 ± 11.4	IIIC-IV EOC, primary peritoneal or fallopian tube cancer	PDS	Low	BMI	− Extent of debulking surgery − Overall survival/mortality rate − Post-surgical complications − Progression-free survival − Toxicity-induced modification of treatment
Kumar 2016 [19]	USA	296	Mean: 64.6 ± 10.6	IIIC-IV EOC	PDS followed by (86.8%) or not followed by (3.4%) chemotherapy, unclear (9.8%)	Low	Muscle density Muscle mass	− Overall survival − Progression-free survival
Lv 2019 [52]	China	362	Mean: 44.78 = ±9.17 only patients aged 35–55 included in analysis	I-IV OC	Surgery	Low	BMI	− Length of hospital stay − Overall survival − Post-surgical complications
Mahdi 2016 [79]	USA	2061	47% = 0–59 28% = 60–69 18% = 70–79 6.8% = ≥80	OC	Surgery	Low	BMI	− Overall survival − Post-surgical complications
Mardas 2017 [80]	Poland	190	Mean: FIGO I-II = 53.8 ± 9.9 FIGO III-IV = 57.5 + 11.5	I-II (28.9%), III-IV (71.1%) EOC	PDS followed by chemotherapy (86.3%), NACT-IDS (13.7%)	Low	Weight and change	− Overall survival − Progression-free survival
Matsubara 2019 [81]	Japan	92	Mean: 55.3 (15–78)	I-IV OC	PDS followed by chemotherapy (66.3%), NACT-IDS (33.7%)	Low	Muscle mass	− Overall survival − Progression-free survival
Matthews 2009 [82]	USA	304	Mean: BMI < 30 = 62.2 ± 11.3 BMI ≥ 30 = 58.3 ± 11.6	II-IV EOC	PDS followed by chemotherapy	Moderate	BMI	− Extent of debulking surgery − Intra-operative outcomes − Length of hospital stay − Overall survival − Platinum sensitivity − Post-surgical complications − Progression-free survival
Munstedt 2008 [83]	Germany	824	Mean: 60.9 ± 13.1	I-IV EOC	Surgery, chemotherapy and/or radiation therapy (NR)	Low	BMI	− Overall survival
Nakayama 2019 [84]	Japan	94	Mean: 61.8 (25–84)	I-IV OC	PDS followed by chemotherapy	Moderate	Muscle density Muscle mass	− Disease-free survival − Overall survival
Orskov 2016 [21]	Denmark	2654 (BMI unknown for 3%)	Median: ≤64 = 52% >64 = 48%	I-IV OC, I-II (36%), III-IV 63%), unknown (1%)	Surgery	Low	BMI	− Overall survival
Pavelka 2006 [5]	USA	216	Mean: BMI < 18.5 = 59.8 BMI 18.5–24.9 = 57.3 BMI 25–29.9 = 63.9 BMI ≥ 30 = 59.3	I-IV EOC or primary peritoneal cancer	PDS	Moderate	BMI	− Extent of debulking surgery − Overall survival − Progression-free survival
Pinar 2017 [85]	Turkey	112	Median: 56.4 (20–80)	I-II (17.8%), III-IV (82.2%) EOC	PDS followed by chemotherapy (78.6%) and (9.9%)/or (20.5%) radiation therapy	Low	BMI	− Overall survival
Popovic 2017 [45]	Republic of Srpska	163	Mean: 59.03 ± 11.81	III-IV OC (including non-epithelial OC)	Surgery	Low	BMI	− Overall survival
Previs 2014 [86]	USA	81	Median: 56 (21–86)	I-IV EOC	Surgery	Low	BMI	− Disease-specific survival − Overall survival − Progression-free survival
Roy 2020 [87]	USA	1786	<50 = 311 50–59 = 490 60–69 = 543 ≥70 = 442	OC or primary peritoneal cancer	Surgery	Low	BMI	− Discharge location
Rutten 2016 [88]	Netherlands	123	Mean: 66.5 ± 0.8	IIB-IV OC	NACT-IDS	Low	Fat mass change Muscle mass and change	− Overall survival
Rutten 2017 [89]	Netherlands	216	Mean: 63.1 ± 0.8	II-IV OC	PDS	Low	Fat mass Muscle density Muscle mass	− Overall survival − Post-surgical complications
Schlumbrecht 2011 [90]	USA	194 (BMI unknown for 29.7%)	Mean: 44.9	I-IV EOC	PDS followed by chemotherapy or NACT-IDS, 12.4% received hormone treatment after adjuvant chemotherapy	Low	BMI	− Overall survival − Progression-free survival
Skirnisdottir 2008 [91]	Sweden	635	Mean: 60	IA-IIC EOC	PDS followed by chemotherapy (47.7%) or radiotherapy (52.3%)	Low	BMI	− Disease-specific survival − Overall survival − Progression-free survival
Skirnisdottir 2010 [92]	Sweden	446	Mean: 62.5 (25–91)	I-II (36%), III-IV (64%) EOC	PDS followed by chemotherapy	Low	BMI	− Disease-specific survival − Overall survival
Slaughter 2014 [93]	USA	46	Median: PDS group = 62.4 PDS + BEV group = 63.4	III-IV EOC	PDS followed by chemotherapy (N = 25) or PDS followed by chemotherapy with BEV (n = 21)	Low	BMI Fat mass	− Overall survival − Progression-free survival
Smits 2015 [94]	UK	228	Median: BMI < 25 = 63.1 (21–88) BMI 25–29.9 = 65.6 (28–85) BMI ≥ 30 = 64.6 (19–81)	I-IV OC, primary peritoneal or fallopian tube cancer	PDS followed by chemotherapy (82%) or NACT-IDS (28%)	Low	BMI	− Extent of debulking surgery − Intra-operative outcomes − Length of hospital stay − Overall survival − Post-surgical complications − Re-hospitalization
Son 2018 [95]	UK	68	Median: 57 (38–80)	IIIC-IVB EOC	NACT-IDS	Moderate	BMI	− Extent of debulking surgery
Staley 2020 [96]	USA	201	Median: 63.6 (24.1–91.5)	I-IV EOC	PDS followed by chemotherapy, NACT-IDS (NR)	Moderate	Muscle mass	− Chemotherapy toxicity − Overall survival − Progression-free survival − Toxicity-induced modification of treatment − Treatment-related hospitalizations
Suh 2012 [53]	Korea	486	Mean: BMI < 23.0 = 48.6 BMI ≥ 23.0 = 53.2	I-IV EOC or primary peritoneal cancer I-II (36.6%), III-IV (62.6%), unknown (0.8%)	PDS followed by chemotherapy, NACT-IDS (9.3%)	Low	BMI	− Extent of debulking surgery − Intra-operative outcomes − Length of hospital stay − Overall survival − Platinum sensitivity − Post-surgical complications − Progression-free survival
Torres 2013 [97]	USA	82	Mean: 67.4 ± 11.7	IIIC-IV OC	PDS	Low	BMI Fat mass Muscle mass	− Length of hospital stay − Overall survival − Post-surgical complications
Ubachs 2020 [46]	Netherlands	212	Mean: 60.9 ± 8.2	III EOC, primary peritoneal or fallopian tube cancer	NACT	Moderate	Muscle mass change	− Chemotherapy toxicity − Overall survival − Recurrence-free survival
Uccella 2018 [7]	Italy	70 (52 included in analysis on post-surgical complications	Median: 58.5 (27–78)	IIIC-IV OC	PDS	Low	BMI	− Extent of debulking surgery − Post-surgical complications
Vitarello 2021 [98]	USA	102	Median: 64 (38–90)	III-IV OC	NACT	Moderate	BMI Fat mass Muscle mass	− Extent of debulking surgery
Wade 2019 [99]	USA	1538	3.4% = <40 14.6% = 40–49 32.3% = 50–59 32.2% = 60–69 15.6% = 70–79 1.8% = ≥80	III-IV EOC, primary peritoneal or fallopian tube cancer	PDS followed by chemotherapy with or without BEV (NR)	Moderate	BMI Fat mass	− Overall survival
Wang 2021 [100]	China	273 (BMI unknown for 7.3%)	Median (IQR): 51 (46–60)	IIIC-IV EOC	PDS followed by chemotherapy (35.6%), NACT (64.4%)	Low	BMI	− Overall survival − Progression-free survival
Wolfberg 2004 [101]	USA	128	Mean (SE): BMI < 30 = 56.3 (1.26) BMI ≥ 30 = 55.7 (2.11)	III-IV EOC	Surgery	Moderate	BMI	− Extent of debulking surgery − ICU admission − Length of hospital stay − Post-surgical complications
Wright 2008 [102]	USA	387	Median: 56.8 (21.8–85.5)	III EOC	PDS followed by chemotherapy	Low	BMI	− Chemotherapy toxicity − Overall survival − Progression-free survival − Toxicity-induced modification of treatment
Yan 2021 [103]	China	415	Median: 50 (25–75)	III-IV EOC	PDS incorporating bowel resection	Low	BMI	− Overall survival − Progression-free survival
Yao 2019 [104]	USA	535	Mean: 64.3 ± 11.3	IIIC-IV EOC, primary peritoneal or fallopian tube cancer	PDS followed by chemotherapy	Low	BMI	− Discharge location − ICU-admission
Yim 2016 [10]	Korea	213	Median: 53 (22–81)	III-IV EOC	PDS followed by chemotherapy	Low	BMI	− Overall survival − Progression-free survival
Yoshikawa 2017 [105]	Japan	76	Median: 62 (33–81)	I-IV OC	Chemotherapy	Low	Muscle mass	− Chemotherapy toxicity
Yoshikawa 2021 [106]	Japan	72	Median: High psoas muscle index = 60 (33–78) Low psoas muscle index = 65 (41–81)	I-IV EOC	PDS followed by chemotherapy (N = 41), NACT-IDS (N = 31)	Low	Muscle mass	− Overall survival
Yoshino 2020 [54]	Japan	60	Median: 63.5 (43–81)	III-IV EOC	Induction chemotherapy	Low	BMI Muscle mass and change	− Overall survival
Zanden, van der 2021 [107]	Netherlands	213	Median: 75.9 (70–89)	IIIA-IV OC	Surgery	Low	Muscle density Muscle mass	− Discharge location − Length of hospital stay − Post-surgical complications − Re-hospitalization
Zhang 2004 [55]	China	254	Alive = 44.1 ± 13.7 Deceased = 51.1 ± 9.0	I-IV EOC	NR	Low	Green tea consumption	− Overall survival
Zhang 2005 [44]	China	207	Alive = 46.7 ± 12.7 Deceased = 51.6 ± 8.8	I-IV EOC	Surgery and chemotherapy	Low	BMI	− Overall survival
**Experimental studies**								
**Author** **Year** **Country**	**Study design**	**Sample size**	**Age (years) ( ± SD or range)**	**FIGO stage (% of patients)**	**Treatment (% of patients)**	**Risk of bias assessment**	**Intervention (duration and frequency) versus comparison**	**Outcome**
Newton 2011 Australia [108]	Non-randomized phase 2 trial	17	Mean: 60.4 (44–71)	I-IV EOC (76%) or primary peritoneal cancer (24%)	PDS followed by chemotherapy (82%) or chemotherapy followed by IDS (18%)	High	Weekly individualized walking prescription by an exercise physiologist, supervised biweekly (in-person or telephone) meetings	− Anxiety − Depression − Ovarian-specific concerns − Physical symptoms − Quality of life − Six-minute walk test
Qin 2021 China [109]	Randomized controlled trial	60	Mean: 53.3 (10.32) intervention group and 54.67 (11.91) control group	I-IV OC	Completed primary treatment and decided to receive chemotherapy treatment	High	Nutrition education by a nutritionist and 250 mL oral nutrition supplements (1.06 kcal, 0.0356 g protein/mL) three times a day versus nutrition education alone	− Biochemical tests − Nutritional risk
Von Gruenigen 2011 USA [110]	Prospective, single group trial	27	Mean: 59.6 ± 9.2 (45–76)	I-IV EOC, primary peritoneal or fallopian tube cancer	Receiving at least 6 cycles of adjuvant chemotherapy	High	1 guided session every chemotherapy visit for 6 cycles. Individual sessions by registered dietitian. Guidance on intake of nutrient-dense food and staying as physically active as possible	− Dietary intake − Physical activity − Quality of life − Symptoms
Zhang 2018 China [111]	Randomized, single-blind controlled trial	67	Range 18–65 with ~45% in the range of 46–55 years	I-V OC	Surgery and completed first cycle of adjuvant chemotherapy	High	Nurse-led, home-based exercise and cognitive behavioral therapy versus usual care	− Cancer-related fatigue − Depression − Sleep quality − Total fatigue

All studies which examine body composition measures (i.e., muscle mass, muscle density and/or fat mass) used computed tomography scans. Abbreviations: BEV, bevacizumab; BMI, body mass index; (E)OC, (epithelial) ovarian cancer; FIGO, International Federation of Gynaecology and Obstetrics; ICU, intensive care unit; IDS, interval debulking surgery; NACT, neoadjuvant chemotherapy; NR, not reported; PDS, primary debulking surgery; SD, standard deviation; SE, standard error; VO_2_ max, the volume of oxygen the body uses during exercise.

#### 3.2.1. Associations between Energy Balance-Related Factors or Behaviors at Diagnosis and Survival

The best-evidence synthesis provided strong evidence that BMI was not significantly associated with overall survival (OS, *n* = 37), progression-free survival (PFS, *n* = 24), disease-specific survival (*n* = 3), or recurrence-free survival (*n* = 3, Table 5). The meta-analyses also demonstrated no significant association between BMI and OS (*n* = 14, HR: 1.07, 95% CI: 0.88; 1.30, *p* = 0.480, Table 6, Figure 2A). We found no significant differences between subgroups with different BMI classifications (test for subgroup difference: Chi-Square = 3.24, I^2^ = 69%, *p* = 0.074). Neither associations observed for studies using a BMI cut-off of <30 kg/m^2^ (*n* = 8, HR: 0.88, 95%CI: 0.65; 1.19, I^2^ = 38%, *p* = 0.412), nor for studies using a BMI cut-off of ≥30 kg/m^2^ (*n* = 6, HR: 1.28, 95% CI: 0.97; 1.68, I^2^ = 79%, *p* = 0.084) were statistically significant. In addition, no significant association was observed between BMI and PFS (*n* = 8, HR: 1.11, 95% CI: 0.89; 1.38, *p* = 0.350, Table 6, Figure 3A). Outliers were not identified. Publication bias was not observed for the association between BMI and OS (Figure 4, intercept = 0.034, τ = 0.057, *p* = 0.955). 

The best-evidence synthesis showed strong evidence that muscle mass (measured with SMI) was not significantly associated with OS (*n* = 17) or PFS (*n* = 8). In contrast, the meta-analyses showed a positive association between muscle mass and PFS (*n* = 3, HR: 1.41, 95% CI: 1.04; 1.91, *p* = 0.030, Table 6, Figure 3B). A positive trend was also shown for OS, but it was not statistically significant (*n* = 5, adjusted HR: 1.27, 95% CI: 0.98; 1.64, *p* = 0.070, Table 6). The study of Chae et al. [66] appeared to be an outlier and was therefore omitted from the analysis, resulting in a reduction in the estimated HR and heterogeneity (Table 6, Figure 2B). 

The best-evidence synthesis showed insufficient evidence of the association between muscle density and OS (*n* = 7). However, the meta-analysis showed a statistically significant positive association (*n* = 3, adjusted HR: 2.12, 95% CI: 1.62; 2.79, *p* < 0.001, Table 6). The study of Kumar et al. [19] was considered an outlier and omitted from the analysis, resulting in an increase in the estimated HR and a reduction in heterogeneity (Table 6, Figure 2C).

There was strong evidence that fat mass was not significantly associated with PFS (n = 4). Finally, there was insufficient evidence of an association between fat mass (*n* = 11), physical fitness (*n* = 1), and diet (*n* = 1) with OS, between muscle mass and disease-free survival (*n* = 2), and between muscle density and both PFS (*n* = 3) and disease-free survival (*n* = 1).

**Table 5 cancers-14-04567-t005:** Association between body mass index or body composition and clinical outcomes (*n* = 71).

Survival Outcomes																
	Body Mass Index				Muscle Mass				Muscle Density				Fat Mass			
	N+	N-	NS	LoE	N+	N-	NS	LoE	N+	N-	NS	LoE	N+	N-	NS	LoE
Overall survival	*n* = 4 ([4,49,69,86]) *	n = 3 [45,52,90]	*n* = 30 [5] †, [10], [21] *, [24] *, [44] *, [50] *, [53,54], [82] †, [94], [59] *, [61], [62] †, [63], [65] *, [71], [72] *, [73,79,80,83], [85] *, [91,92], [93] *^b,d^, [97], [99] †, [100,102,103]	A	*n* = 4 [11], [66] *, [63] *, [106]		*n* = 13 [19], [50] *, [54], [58], [59] *, [64] *, [68], [81], [84] †, [88], [89] *, [96] †, [97]	A	*n* = 4 [19] *, [58] *, [59] *, [64] *		*n* = 3 [11], [84] †, [89]	C1	*n* = 1 [97]	*n* = 2 [50] ^b^, [93] ^a^	*n* = 8 [11], [50] ^c^, [64], [68], [89], [97], [99] †, [93] ^d^	C1
Progression-free survival		*n* = 5 [5] †^e^, [80,90], [93] ^b^, [100]	*n* = 19 [4,10,49], [50] *, [53], [82] †, [59] *, [61] *, [62] †, [65] *, [69], [71] *, [72,76,86,91], [93] *^d^, [102] *, [103]	A	*n* = 1 [11]	*n* = 1 [63] *	*n* = 6 [19], [50] *, [59] *, [68], [81], [96] †	A	*n* = 1 [11]		*n* = 2 [19,59]	C1			*n* = 4 [11], [50] ^a^, [68], [93] ^d^	A
Disease-free survival			*n* = 1 [69]	C2	*n* = 1 [66]		*n* = 1 [84] †	C1			*n* = 1 [84] †	C2				
Platinum disease-free survival			*n* = 1 [69]	C2												
(Platinum) Recurrence-free survival			*n* = 3 [53], [82] †, [76]	A												
Disease-specific survival			*n* = 3 [86,91,92]	A												
	**Change in body mass index/weight**				**Change in muscle mass**				**Change in muscle density**				**Change in fat mass**			
	N+	N-	NS	LoE	N+	N-	NS	LoE	N+	N-	NS	LoE	N+	N-	NS	LoE
Overall survival		*n* = 5 [49,51,57,74,80]		A	*n* = 4 [11], [51] ^f^, [54,88]		*n* = 3 [46], [51] ^g^, [63]	C1			*n* = 1 [11]	C2	*n* = 2 [51] ^g^, [88]		*n* = 2 [11], [51] ^f^	C1
Progression-free survival		*n* = 3 [49,51,80]	*n* = 1 [74]	A	*n* = 1 [11]		*n* = 1 [51]	C1			*n* = 1 [11]	C2			*n* = 2 [11,51]	A
Recurrence-free survival							*n* = 1 [46]	C2								
**Surgical outcomes**																
	**Body mass index**				**Muscle mass**				**Muscle density**				**Fat mass**			
	N+	N-	NS	LoE	N+	N-	NS	LoE	N+	N-	NS	LoE	N+	N-	NS	LoE
Intra-operative outcomes			*n* = 3 [53] ^h,i^, [82] †^h,i,j^, [94] ^h,j^	A												
Total post-surgical complications	*n* = 4 [52], [60] †, [77] *, [78] *		*n* = 11 [4] *, [7,20,53], [82] †, [94], [70] *, [71] *, [79] *, [97], [101] †	C1			*n* = 5 [68,75,89,97,107]	A		*n* = 1 [107]	*n* = 1 [89]	C1	*n* = 1 [75]		*n* = 3 [75,89,97]	C1
Specific post-surgical complications	*n* = 4 [53] ^k^, [82] ^k^, [94] ^k^, [58] ^l^			A						*n* = 1 [107] ^m^		C2				
Discharge location (other than home)	n = 1 [104]		*n* = 1 [87]	C1						*n* = 1 [107]		C2				
Extent of debulking surgery	*n* = 1 [98] †	*n* = 1 [95] †	*n* = 10 [4], [5] †, [7,48,53], [82] †, [94], [62] †, [71], [101] †	A		*n* = 1 [98] †		C2							*n* = 1 [98] †	C2
ICU-admission		*n* = 1 [101] †	*n* = 1 [104]	C1			*n* = 1 [68]	C2								
Length of hospital stay	*n* = 1 [52]		*n* = 5 [53], [82] †, [94,97], [101] †	A			*n* = 2 [68,97]	A			*n* = 1 [107]	C2	*n* = 1 [97]		*n* = 1 [97]	C1
Re-hospitalization	*n* = 2 [18,78]		*n* = 1 [94]	C1							*n* = 1 [107]	C2				
**Chemotherapy outcomes**																
	**Body mass index**				**Muscle mass**				**Muscle density**				**Fat mass**			
	N+	N-	NS	LoE	N+	N-	NS	LoE	N+	N-	NS	LoE	N+	N-	NS	LoE
Response		*n* = 1 [65]	*n* = 1 [69]	C1												
Toxicity induced modification of treatment	*n* = 1 [72] ^n^	*n* = 2 [20] ^o^, [102] ^n,o^	*n* = 5 [4] ^o^, [62] †^n^, [69] ^p^, [78] ^o^, [102] ^p^	C1			*n* = 3 [64], [68] ^o^, [96] †^n,o^	A		*n* = 1 [64]		C2		*n* = 1 [64]		C2
Total toxicities			*n* = 1 [69]	C2			*n* = 4 [64] ^q^, [68], [96] †, [105] ^q^	A		*n* = 1 [64] ^q^		C2		*n* = 1 [64] ^q^		C2
Specific toxicities		*n* = 1 [102] ^r^	*n* = 2 [69] ^r,s^, [102] ^t,u,v^	C1		*n* = 1 [105] ^t,u^	*n* = 2 [96] †^r^, [105] ^r^	C1								
Complications			*n* = 2 [67] †^x^, [69] ^w^	B												
Treatment-related hospitalizations							*n* = 1 [96] †	C2								
	**Change in body mass index/weight**				**Change in muscle mass**				**Change in muscle density**				**Change in fat mass**			
	N+	N-	NS	LoE	N+	N-	NS	LoE	N+	N-	NS	LoE	N+	N-	NS	LoE
Total toxicities						*n* = 1 [46]		C2								

Studies with * are included in meta-analysis and studies with † have a moderate risk of bias (all other studies have a low risk of bias. There are no studies with a high risk of bias.). ^a^ In patients with low skeletal muscle index, ^b^ in bevacizumab group, ^c^ in patients with normal/high skeletal muscle index, ^d^ in chemotherapy group, ^e^ in patients with stage III/IV, ^f^ volumetric muscle mass, ^g^ sectional muscle mass, ^h^ blood loss, ^i^ operating room time, ^j^ transfusion rate, ^k^ wound complications (in BMI > 30 vs. <30 or >40 vs. <40), ^l^ re-operation, ^m^ infectious complications, ^n^ chemotherapy dose intensity, ^o^ time to chemotherapy initiation, ^p^ chemotherapy completion, ^q^ grade ≥ 3 toxicities, ^r^ (grade ≥ 3) hematologic toxicities, ^s^ fatigue, ^t^ grade < 3 events, ^u^ neurologic toxicities, ^v^ gastro-intestinal, genitourinary, or metabolic toxicities, ^w^ catheter malfunction or other complications, ^x^ thromboembolism or infection. Abbreviations: LoE, level of evidence; N+, an increase in determinant is associated with an increase in outcome; N-, an increase in determinant is associated with a decrease in outcome; NS, an increase in determinant is not associated with a statistically significant difference in outcome.

**Table 6 cancers-14-04567-t006:** Meta-analyses of the association between body composition measures and clinical outcomes.

			Main Effect		
Outcomes	*n*	Sample Size	HR (95% CI)	*p*-Value	I^2^
**Overall survival**					
Body mass index					
Overall effect	14	5058	1.07 (0.88; 1.30)	0.480	64%
Skeletal muscle mass					
Overall effect	6	961	1.38 (0.93; 2.03)	0.110	55%
Without outlier ^a^	5	879	1.27 (0.98; 1.64)	0.070	15%
Skeletal muscle density					
Overall effect	4	998	1.80 (1.20; 2.70)	0.004	78%
Without outlier ^b^	3	702	2.12 (1.62; 2.79)	<0.001	0%
**Progression-free survival**					
Body mass index					
Overall effect	8	1350	1.11 (0.89; 1.38)	0.350	45%
Skeletal muscle mass					
Overall effect	3	424	1.41 (1.04; 1.91)	0.030	9%
**Outcome**	n	Sample size	OR (95% CI)	*p*-value	I^2^
**Post-surgical complications**					
Body mass index					
Overall effect	6	3863	1.94 (1.16; 3.24)	0.010	67%
Without outlier ^c^	5	1802	1.63 (1.06; 2.51)	0.030	55%

^a^ Study of Chae et al., 2021 was an outlier [66], ^b^ study of Kumar et al., 2016 was an outlier [19], ^c^ study of Inci et al., 2021 was an outlier [77]. Abbreviations: CI, confidence interval; HR, hazard ratio; I^2^, heterogeneity between studies; n, number of studies included in analysis; OR, odds ratio.

**Figure 2 cancers-14-04567-f002:**
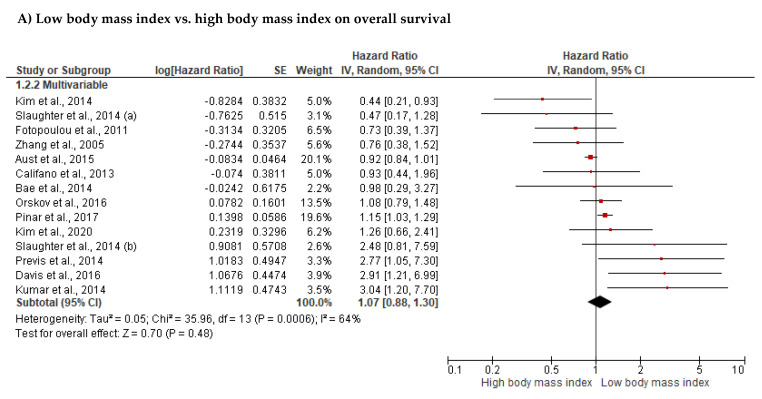
Association of (**A**) body mass index (Kim et al., 2014 [49], Slaughter et al., 2014 [93], Fotopoulou et al., 2011 [71], Zhang et al., 2005 [44], Aust et al., 2015 [59], Califano et al., 2013 [65], Bae et al., 2014 [24], Orskov et al., 2016 [21], Pinar et al., 2017 [85], Kim et al., 2020 [50], Previs et al., 2014 [86], Davis et al., 2016 [69], Kumar et al., 2014 [4]), (**B**) muscle mass (Chae et al., 2021 [66], Bronger et al., 2016 [63], Rutten et al., 2017 [89], Aust et al., 2015 [59], Bruno et al., 2021 [64], Kim et al., 2020 [50]) and (**C**) muscle density with overall survival Bruno et al., 2021 [64], Aust et al., 2015 [59], Ataseven et al., 2018 [58], Kumar et al., 2016 [19].

**Figure 3 cancers-14-04567-f003:**
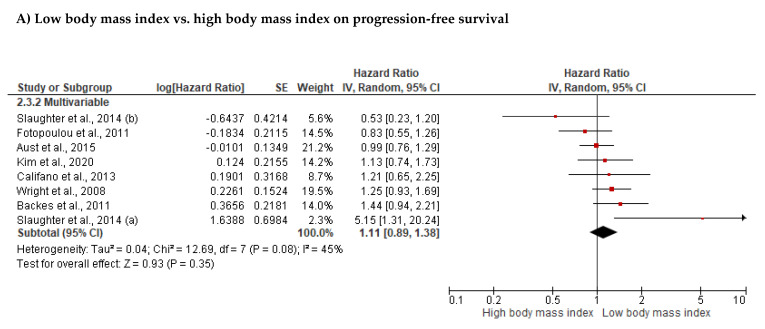
Association of (**A**) body mass index (Slaughter et al., 2014 [93], Fotopoulou et al., 2011 [71], Aust et al., 2015 [59], Kim et al., 2020 [50], Califano et al., 2013 [65], Wright et al., 2008 [102], Backes et al., 2011 [61]) and (**B**) muscle mass with progression-free survival (Bronger et al., 2016 [63], Aust et al., 2015 [59], Kim et al., 2020 [50]).

**Figure 4 cancers-14-04567-f004:**
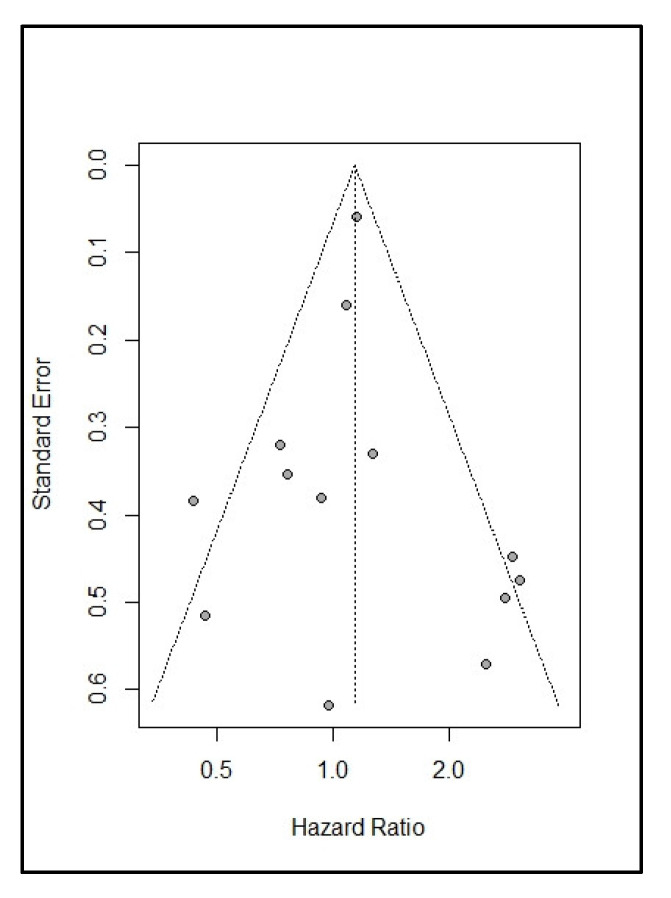
Contour-enhanced funnel plot for the association of body mass index with overall survival.

#### 3.2.2. Associations between Body Weight or Body Composition Changes during Treatment and Survival

There was strong evidence that a reduction in body weight was significantly associated with a shorter OS (*n* = 5) and PFS (*n* = 4, Table 5). In addition, there was strong evidence that a change in fat mass was not associated with PFS (*n* = 2). There was insufficient evidence of associations between a change in muscle mass and OS (*n* = 7) or PFS (*n* = 2), between a change in fat mass and OS (*n* = 4), between a change in muscle mass and recurrence-free survival (*n* = 1), and between a change in muscle density and OS (*n* = 1) and PFS (*n* = 1).

#### 3.2.3. Associations between Body Composition and Surgical Outcomes

The best-evidence synthesis showed strong evidence that BMI was not significantly associated with intra-operative outcomes (*n* = 3), the extent of cytoreductive surgery (*n* = 12), or length of hospital stay (LOS, *n* = 6, Table 5). There was insufficient evidence for any association between BMI and post-surgical complications (*n* = 15). However, our meta-analysis revealed that a higher BMI was significantly associated with a higher risk of developing post-surgical complications (*n* = 5, adjusted OR: 1.63, 95% CI: 1.06; 2.51, *p* = 0.030, Figure 5). The study of Inci et al. [77] was considered an outlier and omitted from the analysis, resulting in a decrease in the estimated OR and heterogeneity (Table 6). Additionally, there was strong evidence that a higher BMI was significantly associated with more wound complications (*n* = 3) and that there was no association between muscle mass and LOS (*n* = 2) or post-surgical complications (*n* = 5).

There was insufficient evidence for other associations between body composition measures and surgical outcomes (Table 5).

#### 3.2.4. Associations between Body Composition and Chemotherapy Outcomes 

The best-evidence synthesis provided strong evidence that muscle mass was not significantly associated with total toxicities (*n* = 4) and toxicity-induced modifications of treatment (*n* = 3), and moderate evidence that BMI was not significantly associated with chemotherapy-related complications (*n* = 2, Table 5). There was insufficient evidence for other associations between body composition and chemotherapy outcomes.

### 3.3. Experimental Studies

Two studies [108,111] examined the effect of an exercise intervention, one study [61] examined a dietary intervention, and another study [110] examined a combined exercise and dietary intervention (Table 3). All experimental studies had a high risk of bias (Table 4). 

Table 7 summarizes the results of the experimental studies. One randomized controlled trial (RCT) showed a potential beneficial effect of exercise on fatigue, depression, and sleep quality [111]. Another exercise trial showed improvements in the six-minute walk test, but not for quality of life, anxiety, or depression scores [108]. One RCT showed a potential beneficial effect of magnesium supplementation on renal function [109]. Analysis of within-group data showed beneficial effects of an exercise and diet intervention on quality of life and symptom scores [110].

## 4. Discussion

This review and meta-analysis synthesized current evidence from observational studies on the association between energy-balance related factors or behaviors and clinical outcomes in patients with ovarian cancer. Additionally, we synthesized the current evidence from experimental studies focusing on exercise and diet during treatment. There were three main findings. First, BMI at diagnosis was not significantly associated with survival outcomes. Second, we found preliminary indications that a higher muscle mass and density were associated with better survival outcomes, but not with surgical outcomes or toxicity. Finally, both observational and experimental studies focusing on exercise, sedentary behavior, and diet are limited.

Findings from previous reviews examining the association between BMI and survival in patients with ovarian or other types of cancer were conflicting, reporting positive, negative, or no significant associations [12,25,112,113]. Our study clearly showed no association between BMI and survival, indicating that BMI at ovarian cancer diagnosis has a limited prognostic value. This may be due to disease-specific symptoms such as ascites influencing body weight, or due to BMI not adequately reflecting fat and muscle mass proportions. In line with this, our meta-analyses showed that muscle mass and density may have prognostic value for OS and PFS. This supports previous findings in patients with other cancer types [114,115,116,117], and skeletal muscle has been recognized as an endocrine organ, secreting myokines and other factors that may help to control tumor growth [118]. In addition, previous studies have shown that behavioral interventions, such as resistance exercise and/or a sufficient protein intake, may positively influence muscle mass [117,119,120,121]. 

However, the results regarding the association between muscle mass and density and survival outcomes differed between the meta-analyses and the best-evidence syntheses. In both cases, the best-evidence syntheses incorporated a larger number of studies with inconsistent findings. This suggests that the results of the meta-analyses may have been affected by reporting bias, due to studies not reporting sufficient information to be included in the analysis. This is particularly problematic in situations where individual studies may have had a lack of power to detect a statistically significant association. Unfortunately, we were not able to examine publication bias in all meta-analyses, as at least ten studies had to be included for these analyses to be valid. Future studies should appropriately report point estimates and measures of variability on all outcomes. This would improve the interpretability of the outcomes and allow for inclusion in future meta-analyses to clarify their prognostic value. 

Similarly, although the best-evidence synthesis yielded insufficient evidence, the results of the meta-analyses were that a higher BMI was significantly associated with an increased risk of post-operative complications. Particularly, BMI was associated with specific problems such as wound complications [53,82,94]. The higher rate of wound complications in patients with a higher BMI, and especially those with morbid obesity, may be explained by a higher fat mass. This may be due to vascular insufficiencies, systemic inflammation, oxidative stress, or nutritional deficiencies, resulting in weakened immune function and compromised recovery [122]. There were only a few studies available; thus, more evidence is needed to clarify the association between fat mass and surgical complications. 

Besides muscle mass, showing no associations, there is generally insufficient evidence on the association between body composition and chemotherapy-related outcomes. A previous study presented that the clearance of cisplatin and paclitaxel was increased in obese patients [123]. However, underlying mechanisms for the effect of obesity on treatment outcome are currently unknown [123], and a study in patients receiving paclitaxel for esophageal cancer reported that paclitaxel dosing could not be optimized by correcting for body composition [124]. Future studies should identify if body composition measures have prognostic value for specific toxicities in patients with ovarian cancer. 

Our recommendation is that we need to move beyond BMI in order to assess body composition as a prognostic variable. The studies included in our review generally determined muscle mass and density using CT scans routinely collected in clinical practice, allowing valid and reliable measures of fat and muscle mass and muscle quality [125,126]. However, the analyses are currently time consuming. Rapidly evolving technological innovations hold promise to achieve automatic body composition analyses of CT scans. Additionally, understanding the prognostic value of other measures of muscle mass, muscle density, and fat mass, including a multifrequency bioelectrical impedance analysis, which can adjust for ascites [127], dual energy X-ray absorptiometry, or ultrasound are needed to inform the design and implementation of ovarian cancer-specific exercise and/or dietary interventions in clinical settings.

The strengths of this review and meta-analyses are the comprehensive assessment of various body composition measures and survival and treatment-related outcomes, and the focus on energy balance-related behavioral interventions, specifically in patients with ovarian cancer. However, our findings are limited by the substantial heterogeneity in the measurements and cut-off values for muscle and fat measures utilized by the included studies. Additionally, the observational design of the studies limits the inferences that can be made on causality. Together with the limited number of experimental studies identified, our review highlights the need for intervention research addressing energy balance-related factors and behavior.

## 5. Conclusions

In this comprehensive review and meta-analysis, we showed that the prognostic value of baseline BMI for clinical outcomes is limited, and that muscle mass and muscle density may have more prognostic potential. More high-quality studies are needed to better understand the prognostic value of muscle and fat measures and energy balance-related behaviors in relation to clinical outcomes, and to determine the effectiveness of interventions targeting energy-balance factors and behaviors in this understudied group of patients with ovarian cancer.

## Figures and Tables

**Figure 1 cancers-14-04567-f001:**
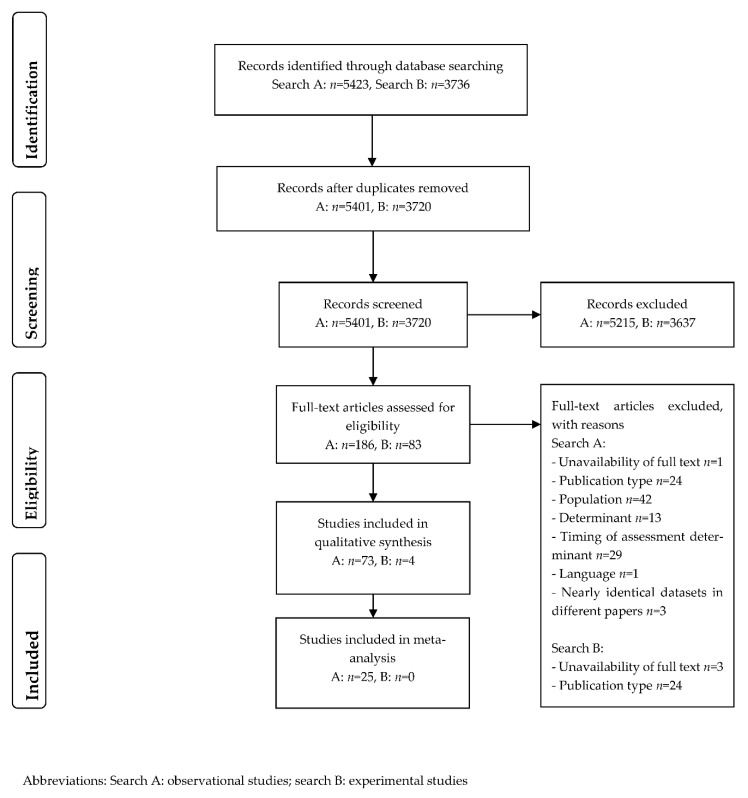
Flow diagram of study selection process.

**Figure 5 cancers-14-04567-f005:**
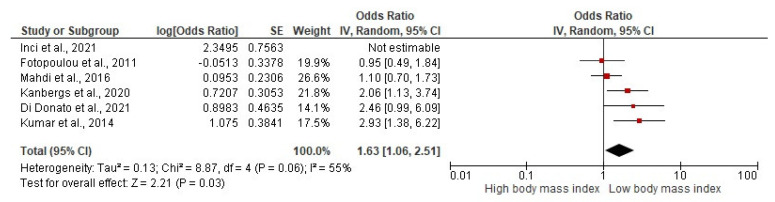
Low body mass index vs. high body mass index on post-surgical complications. Inci et al., 2021 [77], Fotopoulou et al., 2011 [71], Mahdi et al., 2016 [79], Kanbergs et al., 2020 [78], Di Donato et al., 2021 [70], Kumar et al., 2014 [4].

**Table 1 cancers-14-04567-t001:** Overview of inclusion and exclusion criteria.

	Systematic searches			
	Q1: What is the association between body weight, body composition, diet, exercise, sedentary behavior, and physical fitness at diagnosis and during treatment with clinical outcomes in patients with ovarian cancer?		Q2: What is the effect of exercise and/or dietary intervention during treatment in patients with ovarian cancer?	
	**Inclusion**	**Exclusion**	**Inclusion**	**Exclusion**
Availability of full text and language	Full text available (no restriction on publication date); papers written in English	Unavailable full text; non-English language studies	Full text available (no restriction on publication date); papers written in English	Unavailable full text; non-English language studies
Publication type	Original research article	Review, conference abstract, case presentation, commentaries, editorials, grey literature	Original research article	Review, conference abstract, case presentation, commentaries, editorials, grey literature
Population	Studies involving patients with primary epithelial ovarian, peritoneal, or fallopian tube cancer (≥75% of the study sample), or separate reporting of results for patients with epithelial ovarian cancer in studies involving various types of gynecological cancer	Studies involving patients with recurrent or any other type of cancer besides epithelial ovarian, peritoneal or fallopian tube cancer	Studies involving patients with primary epithelial ovarian, peritoneal, or fallopian tube cancer (≥75% of the study sample), or separate reporting of results for patients with epithelial ovarian cancer in a sample of various types of gynecological cancer	Studies involving patients with recurrent or any other type of cancer besides epithelial ovarian, peritoneal, or fallopian tube cancer
Study design	Prospective or retrospective cohort studies, cross sectional studies, case-control studies	Experimental studies	Controlled intervention studies with an attention control, wait-list, or usual care group, randomized controlled trials, non-randomized controlled trials (including pilot studies)	Observational studies
Exposure/intervention	Body weight, body composition, diet, exercise, sedentary behavior, or physical fitness	Mind-body therapies (e.g., yoga, Tai chi), phytochemicals (e.g., carotenoids, flavonoids), or enteral/parenteral nutrition	Exercise and/or nutritional interventions	Mind-body therapies (e.g., yoga, Tai chi), phytochemicals (e.g., carotenoids, flavonoids), or enteral/parenteral nutrition
Timing of assessment of determinant/timing of intervention	At diagnosis and/or during first-line cancer treatment	Before diagnosis or during treatment for recurrent cancer	At diagnosis and/or during first-line cancer treatment	Before diagnosis or during treatment for recurrent cancer
Outcome variable	Treatment-related outcomes (i.e., surgical and chemotherapy-related outcomes) and survival outcomes	All other outcomes	Body weight, body composition, dietary intake, physical activity, biomarkers, patient-reported outcomes (e.g., quality of life, symptoms of ovarian cancer), treatment-related outcomes or survival outcomes	All other outcomes

Abbreviations: BMI, body mass index; Q, research question.

**Table 2 cancers-14-04567-t002:** Example of literature search as conducted in MEDLINE.

Search	Query	Items Found
#41	Search (#38 NOT (animals [mh] NOT humans [mh]))	1874
#39	Search (#37 NOT (animals [mh] NOT humans [mh]))	3266
#38	Search (#31 OR #35)	2061
#37	Search (#31 OR #32 OR #33 OR #34)	3547
#31	Search #25 #26	608
#35	Search #25 #30	1605
#34	Search #25 #29	3066
#33	Search #25 #28	92
#32	Search #25 #27	62
#30	Search (“Nutritional Status”[Mesh] OR “Nutrition Therapy”[Mesh] OR diet[tiab] OR diets[tiab] OR dietary[tiab] OR dietetic*[tiab] OR nutriti*[tiab])	740,947
#29	Search (“Body Composition”[Mesh] OR “Body Fat Distribution”[Mesh] OR “Body Mass Index”[Mesh] OR “Body Weight”[Mesh] OR “Waist Circumference”[Mesh] OR “Waist-Height Ratio”[Mesh] OR “Skinfold Thickness”[Mesh] AND “Waist-Hip Ratio”[Mesh] OR body composition*[tiab] OR body fat*[tiab] OR adiposity[tiab] OR fat mass*[tiab] OR body mass*[tiab] OR muscle mass*[tiab] OR sarcopenia[tiab] OR sarcopaenia[tiab] OR bmi[tiab] OR bmis[tiab] OR waist to hip[tiab] OR waist hip[tiab] OR obese[tiab] OR obesity[tiab] OR body weight*[tiab] OR weight los*[tiab] OR weight gain*[tiab] OR overweight[tiab] OR overweightness[tiab] OR anthropometric*[tiab] OR skeletal muscle index[tiab] OR hip circumference*[tiab] OR waist circumference*[tiab] OR thigh circumference*[tiab] OR abdominal circumference*[tiab] OR skinfold thickness*[tiab] OR fat free mass*[tiab] OR hip waist[tiab] OR hip to waist[tiab])	767,972
#28	Search (“Physical Fitness”[Mesh] OR “Physical Endurance”[Mesh] OR physical fitness[tiab] OR physical function*[tiab] OR cardiorespiratory fitness[tiab] OR physical endurance[tiab] OR physical performance[tiab])	89,758
#27	Search (“Sedentary Behavior”[Mesh] OR sedentary[tiab] OR physical inactivity[tiab] OR physically inactive[tiab])	39,207
#26	Search (“Exercise”[Mesh:noexp] OR “Physical Conditioning, Human”[Mesh] OR “Running”[Mesh] OR “Swimming”[Mesh] OR “Walking”[Mesh] OR “Exercise Therapy”[Mesh] OR exercis*[tiab] OR physical training[tiab] OR endurance training[tiab] OR aerobic training[tiab] OR resistance training[tiab] OR anaerobic training[tiab] OR circuit training[tiab] OR high intensity interval training[tiab] OR hiit[tiab] OR walking[tiab] OR jogging[tiab] OR swimming[tiab] OR running[tiab] OR bicycling[tiab] OR physical activit*[tiab] OR sports activit*[tiab] OR activity behavi*[tiab])	558,674
#25	Search ((“Ovarian Neoplasms”[Mesh] OR ((ovarian[tiab] OR ovary[tiab] OR ovaries[tiab]) AND (neoplasm*[tiab] OR cancer*[tiab] OR tumor[tiab] OR tumors[tiab] OR tumour[tiab] OR tumours[tiab] OR carcinoma*[tiab] OR malignan*[tiab] OR oncolog*[tiab])) OR gynecological cancer*[tiab] OR gynaecological cancer*[tiab]) NOT (polycystic[ti] OR pcos[ti]))	127,070

**Table 4 cancers-14-04567-t004:** Risk of bias assessment of observational and experimental studies.

Observational Studies											
Author, year	Similar groups and recruited from same population?	Exposure measured similarly?	Exposure measured in valid and reliable way?	Confounding factors identified? ^1^	Strategies to deal with confounders stated?	Free of outcome at the start of study?	Outcomes measured in valid and reliable way?	Follow-up time reported and sufficient? ^2^	Follow-up complete? Were reasons to loss to follow-up described and explored? ^3^	Strategies to address incomplete follow-up utilized? ^4^	Appropriate statistical analysis?
Ansell, 1993 [57]	Low	Low	Unclear	Low	Low	Low	Low	Low	Unclear	Unclear	Low
Ataseven, 2018 [58]	Low	Low	Low	High	Low	Low	Low	Low	Unclear	Unclear	Low
Aust, 2015 [59]	Low	Low	Low	Low	Low	Low	Low	Low	Unclear	Unclear	Low
Bacalbasa, 2020 [60]	Low	Unclear	Unclear	High	NA	Low	Low	Low	Low	NA	Unclear
Backes, 2011 [61]	Low	Low	Low	Low	Low	Low	Low	High	Unclear	Unclear	Low
Bae, 2014 [24]	Low	Low	Low	Low	Low	Low	Low	High	Unclear	Unclear	Low
Barrett, 2008 [62]	Low	Low	Low	High	NA	Low	Unclear	High	Unclear	Unclear	Low
Bronger, 2017 [63]	Low	Low	Low	Low	Low	Low	Unclear	Low	Low	Unclear	Low
Bruno, 2021 [64]	Low	Low	Low	Low	Low	Low	Low	Low	Unclear	Unclear	Low
Califano, 2013 [65]	Low	Low	Low	High	Low	Low	Unclear	Low	Unclear	Unclear	Low
Castro, 2018 [20]	Low	Low	Unclear	Low	Low	Low	Low	Low	Low	NA	Low
Chae, 2021 [66]	Low	Low	Low	High	NA	Low	Low	Low	Unclear	Unclear	Low
Chokshi, 2022 [67]	Low	Unclear	Unclear	High	NA	Low	Low	Low	Low	NA	Low
Conrad, 2018 [68]	Low	Low	Low	Low	Low	Low	Low	Low	Unclear	Unclear	Low
Davis, 2016 [69]	Low	Low	Low	Low	Low	Low	Low	High	Unclear	Unclear	Low
Di Donato, 2021 [70]	Low	Low	Unclear	Low	Low	Low	Low	Low	Low	NA	Low
Duska, 2015 [18]	Low	Low	High	Low	Low	Low	Low	Low	Unclear	Unclear	Low
Element, 2022 [56]	Low	Low	Low	High	NA	Low	Low	Low	Low	NA	High
Fotopoulou, 2011 [71]	Low	Low	Low	Low	Low	Low	Unclear	High	Unclear	Unclear	Low
Hanna, 2013 [72]	Low	Low	Unclear	Low	Low	Low	Unclear	Low	Unclear	Unclear	Low
Hawarden, 2021 [73]	Low	Low	Low	High	NA	Low	Low	Low	Low	NA	High
Hess, 2007 [74]	Low	Low	Low	Low	Low	Low	Unclear	High	Unclear	Unclear	Low
Heus, 2021 [75]	Low	Low	Low	Low	Low	Low	Low	Low	Low	NA	Low
Hew, 2014 [76]	Low	Low	Low	Low	Low	Low	Low	High	Low	NA	Low
Huang, 2020 [11]	Low	Low	Low	Low	Low	Low	Low	Low	Unclear	Unclear	Low
Inci, 2021 [77]	Low	Low	Unclear	Low	Low	Low	Low	Low	Low	NA	Low
Jiang, 2019 [48]	Low	Low	Low	Low	Low	Low	Low	Low	Low	NA	Low
Kanbergs, 2020 [78]	Low	Low	Low	Low	High	Low	Low	Low	Low	NA	Low
Kim, 2014 [49]	Low	Low	Low	Low	Low	Low	Low	High	Unclear	Unclear	Low
Kim, 2020 [50]	Low	Low	Low	Low	Low	Low	Low	Low	Unclear	Unclear	Low
Kim, 2021 [51]	Low	Low	Low	High	Low	Low	Low	Low	Low	NA	Low
Kumar, 2014 [4]	Low	Low	Low	Low	Low	Low	Unclear	High	Unclear	Unclear	Low
Kumar, 2016 [19]	Low	Low	Low	Low	Low	Low	Unclear	Unclear	Unclear	Unclear	Low
Lv, 2019 [52]	Low	Low	Unclear	High	NA	Low	Low	Low	Low	NA	Low
Mahdi, 2016 [79]	Low	Low	Unclear	Low	Low	Low	Low	Low	Low	NA	Low
Mardas, 2017 [80]	Low	Low	Low	Low	Low	Low	Low	Low	Unclear	Unclear	Low
Matsubara, 2019 [81]	Low	Low	Low	Low	Low	Low	Unclear	High	Unclear	Unclear	Low
Matthews, 2009 [82]	Low	Low	Unclear	Low	High	Low	Unclear	High	Unclear	Unclear	Low
Munstedt, 2008 [83]	Low	Low	Low	Low	High	Low	Unclear	Low	Low	NA	Low
Nakayama, 2019 [84]	Low	Low	Low	High	NA	Low	Unclear	High	Unclear	Unclear	Low
Orskov, 2016 [21]	Low	Low	Low	Low	Low	Low	Low	Low	Low	NA	Low
Pavelka, 2006 [5]	Low	Low	Low	Low	Unclear	Low	Unclear	High	Unclear	Unclear	Low
Pinar, 2017 [85]	Low	Low	Low	Low	Low	Low	Low	Low	Low	NA	Low
Popovic, 2017 [45]	Low	Low	Low	High	Low	Low	Unclear	Low	High	Unclear	Low
Previs, 2014 [86]	Low	Low	Low	High	Low	Low	Low	High	High	Low	Low
Roy, 2020 [87]	Low	Low	Unclear	Low	Low	Low	Low	Low	Low	Low	Low
Rutten, 2016 [88]	Low	Low	Low	Low	Low	Low	Unclear	High	Unclear	Unclear	Low
Rutten, 2017 [89]	Low	Low	Low	Low	Low	Low	Low	High	Unclear	Unclear	Low
Schlumbrecht, 2011 [90]	Low	Low	Low	Low	Low	Low	Low	Low	Unclear	Unclear	Low
Skirnisdottir, 2008 [91]	Low	Low	Low	High	Low	Low	Unclear	Low	Unclear	Unclear	Low
Skirnisdottir, 2010 [92]	Low	Low	Low	High	Low	Low	Low	Low	Unclear	Unclear	Low
Slaughter, 2014 [93]	Low	Low	Low	Low	Low	Low	Low	High	Unclear	Unclear	Low
Smits, 2015 [94]	Low	Low	Low	Low	High	Low	Low	Low	Low	NA	Low
Son, 2018 [95]	Low	Low	Unclear	High	Low	Low	Low	High	Unclear	Unclear	Low
Staley, 2020 [96]	Low	Low	Low	High	NA	Low	Low	High	Unclear	Unclear	Low
Suh, 2012 [53]	Low	Low	Low	Low	High	Low	Low	Low	Unclear	Unclear	Low
Torres, 2013 97]	Low	Low	Low	Low	Low	Low	Low	Low	Low	NA	Low
Ubachs, 2020 [46]	Low	Low	Low	High	NA	Low	Unclear	Low	Unclear	Unclear	Low
Uccella, 2018 [7]	Low	Low	Low	Low	Low	Low	Low	Low	Low	NA	Low
Vitarello, 2021 [98]	Low	Low	Low	High	NA	Low	Low	High	Unclear	Unclear	Low
Wade, 2019 [99]	Low	Low	Low	High	Low	Low	Unclear	High	Unclear	Unclear	Low
Wang, 2021 [100]	Low	Unclear	Unclear	Low	Low	Low	Low	Low	Low	NA	Low
Wolfberg, 2004 [101]	Low	Low	Unclear	High	NA	Low	Low	High	Low	NA	Low
Wright, 2008 [102]	Low	Low	Low	Low	Low	Low	Low	Low	Unclear	Unclear	Low
Yan, 2021 [103]	Low	Low	Low	High	Low	Low	Low	Low	Low	NA	Low
Yao, 2019 [104]	Low	Low	Unclear	Low	Low	Low	Low	Low	Low	NA	Low
Yim, 2016 [10]	Low	Low	Low	Low	Low	Low	Unclear	Low	Unclear	Unclear	Low
Yoshikawa, 2017 [105]	Low	Low	Low	Low	Low	Low	Low	High	Unclear	Unclear	Low
Yoshikawa, 2021 [106]	Low	Low	Low	Low	Low	Low	Low	Low	Unclear	Unclear	Low
Yoshino, 2020 [54]	Low	Low	Low	Low	Low	Low	Low	High	Unclear	Unclear	Low
Zanden, van der,2021 [107]	Low	Low	Low	Low	Low	Low	Low	Low	Low	Low	Low
Zhang, 2004 [55]	Low	Low	Low	Low	Low	Low	Low	Low	Low	NA	Low
Zhang, 2005 [44]	Low	Low	Low	Low	Low	Low	Low	Low	Low	NA	Low
**Experimental studies**											
Author, year	Randomization process		Effect of assignment to intervention		Effect of adhering to intervention		Missing outcome data		Measurement of outcome		Selective reporting
Newton, 2011 [108]	High (single-arm trial)		High		High		Low		Some concerns		Low
Zhang, 2018 [111]	Low		Some concerns		Some concerns		Some concerns		Some concerns		High
Qin, 2021 [109]	Low		High		High		Low		Low		Some concerns
Von Gruenigen, 2011 [110]	High (single-arm trial)		High		High		Low		Some concerns		High

^1^ Minimum set of confounders that had to be identified were optimal debulking/residual disease, stage, and age. ^2^ A minimum follow up time of 30 days for post-surgical outcomes and 2 years for survival outcomes were considered sufficient. ^3^ Follow up was considered complete when less than 20% of the data was indicated as missing or when loss to follow up was clearly described and explored. ^4^ Not applicable when dropout rate was less than 5%. Abbreviations: NA, not applicable.

**Table 7 cancers-14-04567-t007:** Overview of the results of the physical activity and/or dietary intervention studies (*n* = 4).

Author Year	Adherence	Physical Outcomes	Within/Between Group Differences	Psychosocial Outcomes	Within/Between Group Differences
Newton 2011 [108]	Overall group adherence was 90% (range 55–100%). On average women walked four days a week (range 0–7)	Six-minute walk test Physical symptoms	Median (min, max): 332 (266, 356) to 395 m (356, 460), *p* = 0.01 1.06 (0.0, 2.33) to 0.60 (0.06, 2.06), *p* = 0.14	Anxiety	Median (min, max): 4 (1, 15) to 4 (0.16), *p* = 0.63
				Depression	3 (0, 16) to 4 (0, 13), *p* = 016
				Quality of Life^1^	109 (72, 46), to 113 (67, 148), *p* = 0.10
				Ovarian-specific concerns	31 (20, 41) to 36 (21, 44), *p* = 0.44
Zhang 2018 [111]	83.2% at T1, 76.1% at T2 and 73.7% at T3			Cancer-related fatigue	T2: 4.24 (1.40), 4.94 (1.39), *p* = 0.011 T3: 3.90 (1.42), 5.04 (1.41), *p* = 0.002
				Total fatigue ^1^	T2: 45.03 (7.07), 50.34 (5.88), *p* = 0.001 T3: 43.23 (7.07), 50.04 (5.53), *p* < 0.001
				Symptoms of depression	T2: 7.25 (3.36), 8.86 (3.14), *p* = 0.044
				Sleep quality ^1^	T3: 6.29 (2.96), 7.86 (2.91), *p* = 0.032
Qin 2021 [109]	All participants reported that they completed the intervention goal (750 mL of supplements per day)	Nutritional status	Between-group differences at T1 ^2^ −1.17 (−2.23, −0.11), *p* = 0.01		
		Leukocytes	−0.35 (−1.69, 1.00), *p* = 0.61		
		Lymphocytes	0.41 (−0.04, 0.88), *p* = 0.07		
		Red blood cells	0.05 (−0.20, 0.30), *p* = 0.69		
		Hemoglobin	1.83 (−4.48, 8.15), *p* = 0.57		
		Albumin	3.71 (0.75 (0.75, 6.68), *p* = 0.01		
		Total blood protein	5.49 (−0.36, 11.34), *p* = 0.07		
Von Gruenigen 2011 [110]	92%	Physical activity	Baseline 65 (132), #3: 77(112), #6: 138 (197). *p* = 0.582 (baseline to cycle #3), *p* = 0.063 (cycle #3 to #6) and *p* = 0.082 (baseline to #6).	Quality of life	Baseline: 75.4 #3: 77.6, #6: 83.9 (*p* = 0.001 Baseline-#6)
		Dietary intake	NS		
		Symptoms	Baseline: 20.6, #3: 26.6, #6: 17.0 (*p* = 0.013, #3-#6).		

If available, between-group differences are reported as intervention vs. control group. In the case of single-group design, within-group effects are reported. ^1^ For subscales, see full text paper. ^2^ See full text paper for data at 9- and 15-week follow-up. Abbreviations: #, chemo cycle number; NS not significant; T, timepoint.

## Data Availability

Data can be obtained from the corresponding author.

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
