# Peer review of "Association between Energy Balance-Related Factors and Clinical Outcomes in Patients with Ovarian Cancer: A Systematic Review and Meta-Analysis"

_cancers, 2022, doi:10.3390/cancers14194567_

Round 1

Reviewer 1 Report

Association Between Energy Balance-Related Factors and Clinical Outcomes in Patients with Ovarian Cancer: a Systematic Review and Meta-Analysis

This is an interesting study, but some major issues should be concerned for improving the study quality.

Method:

1. The inclusion criteria for this study were not clear. Please revise.

2. With regard to the outcome, the HRs or ORs were extracted from the original studies. Due to the different calculations of HR and OR, how did you analyze the outcome (HR) in the meta-analysis? Is there any method to convert it? Authors should address this calculation in the method.

3. For the outcomes (e.g., muscle mass, fat mass), authors should indicate what tools were used to test body composition in the method or result and Tables. Otherwise, the reader can not understand it. 

Result:

1. The result (Table 3) about the BMI and overall survival, a larger heterogeneity (64%) was observed, although there was no significant association between them. It is recommended to conduct a sub-analysis to examine the reason. For example, the different classifications of BMI may cause differences in results. 

2. Although the sensitivity analysis was used, what's the purpose of this analysis? I can not find the relevant findings in the result. Please explain. 

3. Please provide forest plots and funnel plots for each outcome, which will be consistent with the method. Additionally, the Egger's test results should be displayed in context with intercept value. 

Tables:

The abbreviations in the table should be addressed fully below the table. For example, EOC was displayed in Table 1, there was no EOC meaning explained below in table 1.  Please check all abbreviations clearly. 

Reviewer 2 Report

In this review Stelten S et al have attempted to perform a systematic review and a meta-analysis to assess if there are any correlations between quality of life and treatment outcomes in ovarian cancer patients and their energy balance related issues such as muscle mass, body mass index, physical activity, nourishment status, etc. Their analysis indicates that measures indicative of the body composition could have prognostic value instead of body mass index. Overall, the review article is well-written and the authors have provided a good background information and discussion. The authors have also highlighted the strengths and limitations of the study. Few comments/ suggestions are listed below:

·         Did the authors look at a correlation between ovarian cancer disease recurrence and body mass index and/ or muscle mass? Can these factors help predict patients who might experience a relapse despite initial response to chemotherapy?

·         There is a lot of literature where BMI is not correlative to ovarian cancer progression but is in fact a predictive prognostic marker for endometrial cancer. Did the authors see positive correlation in studies where the patients were diagnosed with ovarian cancer but also had an involvement of the endometrium?

·         Is there a specific reason for including tabulated results as appendices to the manuscript? Please include them as part of the results in the manuscript to improve readability.

Round 2

Reviewer 1 Report

Thanks to the authors for their response. There are no more questions.